# Clinical and Molecular Prediction of Hepatocellular Carcinoma Risk

**DOI:** 10.3390/jcm9123843

**Published:** 2020-11-26

**Authors:** Naoto Kubota, Naoto Fujiwara, Yujin Hoshida

**Affiliations:** Liver Tumor Translational Research Program, Simmons Comprehensive Cancer Center, Division of Digestive and Liver Diseases, Department of Internal Medicine, University of Texas Southwestern Medical Center, 5323 Harry Hines Blvd, Dallas, TX 75390, USA; Naoto.Kubota@UTSouthwestern.edu

**Keywords:** hepatocellular carcinoma, cirrhosis, precision medicine, cancer screening, risk prediction, biomarker

## Abstract

Prediction of hepatocellular carcinoma (HCC) risk becomes increasingly important with recently emerging HCC-predisposing conditions, namely non-alcoholic fatty liver disease and cured hepatitis C virus infection. These etiologies are accompanied with a relatively low HCC incidence rate (~1% per year or less), while affecting a large patient population. Hepatitis B virus infection remains a major HCC risk factor, but a majority of the patients are now on antiviral therapy, which substantially lowers, but does not eliminate, HCC risk. Thus, it is critically important to identify a small subset of patients who have elevated likelihood of developing HCC, to optimize the allocation of limited HCC screening resources to those who need it most and enable cost-effective early HCC diagnosis to prolong patient survival. To date, numerous clinical-variable-based HCC risk scores have been developed for specific clinical contexts defined by liver disease etiology, severity, and other factors. In parallel, various molecular features have been reported as potential HCC risk biomarkers, utilizing both tissue and body-fluid specimens. Deep-learning-based risk modeling is an emerging strategy. Although none of them has been widely incorporated in clinical care of liver disease patients yet, some have been undergoing the process of validation and clinical development. In this review, these risk scores and biomarker candidates are overviewed, and strategic issues in their validation and clinical translation are discussed.

## 1. Introduction

Hepatocellular carcinoma (HCC), the major histological type of liver cancer, is one of the most rapidly increasing causes of cancer-related mortality in the US and the fourth most common cancer death globally [1]. HCC develops in chronically diseased livers with infection of hepatitis viruses (e.g., hepatitis B virus (HBV) and hepatitis C virus (HCV)) and metabolic insults such as alcohol abuse and non-alcoholic fatty liver disease (NAFLD) [2]. These viral and metabolic etiological factors promote progressive liver fibrosis that results in its terminal stage, cirrhosis, which is estimated to affect 1–2% of the global population and be attributed to 1.32 million deaths annually [2,3]. Despite recent development of effective antivirals for HBV and HCV, the risk of HCC cannot be eliminated, especially when advanced liver fibrosis is established [4,5]. There are no therapies to halt disease progression toward HCC in patients with alcohol abuse and NAFLD [2].

Given the substantial risk of HCC and prolonged survival by early HCC detection [6], with curative options available only for tumors diagnosed at an early stage, multiple professional society guidelines recommend regular biannual screening for HCC, using ultrasound with or without serum alpha-fetoprotein (AFP), in all patients with cirrhosis from any etiologies [7,8,9]. However, the recommended screening is not utilized in many cases (utilization rate <25%), and the diagnostic accuracy of ultrasound and AFP is suboptimal [10]; as a consequence, most HCC tumors are diagnosed at late stages, and median survival remains as less than one year [11]. This is in part due to vast size of target population that overwhelms currently available medical resources for the screening. In addition, the guidelines do not account for heterogeneity in HCC risk among the target patient population. This approach leads to over-screening of low-risk patients and under-screening of high-risk patients [11]. Thus, more precise HCC risk prediction in each individual patient will significantly improve the efficacy of the HCC screening by identifying high-risk patients who most need close monitoring for more efficient detection and diagnosis of early stage HCC amenable to curative therapies. Indeed, Markov-model-based simulation analysis showed that individual-risk-based personalized HCC screening (Figure 1) is more cost-effective compared to the current “one-size-fits-all” screening approach, sparing net medical care costs and improving patient survival [12]. HCC risk stratification may inform tailored HCC screening strategy to maximize cost-effectiveness of HCC screening by optimizing intensity of screening tests according to predicted risk [2,13,14]. For example, more frequent HCC screening can be offered to high-risk patients compared to low-risk patients. It may be justifiable to utilize costly but high-performance screening tests such as advanced imaging modalities (e.g., MRI-based examination) and biomarkers (e.g., circulating cell-free methylated DNA and GALAD score) in high-risk patients. Given the limited resources for HCC screening in real-world clinical practice, prioritizing high-risk patients for regular HCC screening will also be a rational approach. This should be a focus of future research in HCC screening and early detection. A modeling-based study showed that this personalized strategy indeed can be a viable approach to enable cost-effective HCC screening [12]. Such individual-risk-based personalized cancer screening has also been sought in other cancer types, including colorectal cancer and breast cancer [15,16]. With the currently recommended HCC screening based on ultrasound in the all-comers setting, alteration of HCC screening frequency did not influence patient outcome in clinical studies [17]. This issue may be carefully revisited based on these experiences when studying new HCC screening modalities with improved performance with consideration about anticipated tumor growth rate. Sparing HCC screening in low-risk patients could substantially mitigate the burden of regularly screening the large patient population at risk. However, such a decision of dropping a subset of patients from regular screening should be carefully made, to minimize risk of late tumor diagnosis, which may incur increased medical-care costs, despite poor prognosis. Furthermore, with the shift of major HCC etiology from communicable viral infection to metabolic disorders accompanied with low HCC incidence rate and disproportionally affecting the communities with low socioeconomic status, outreach effort to the at-risk population will become increasingly important. In this review article, we overview clinical-variable-based HCC risk scores and molecular biomarkers in the literature, some of which are promising candidate tools to enable such risk-based individualized HCC screening.

## 2. Clinical-Variable-Based HCC Risk Scores

Numerous HCC risk scores have been developed, using clinical variables such as age, sex, laboratory tests, and imaging modalities. We summarized the HCC risk scores and biomarkers that were externally validated in independent patient cohort(s) (Table 1). While some scores were developed for specific HCC etiology, some were trained in regional patients with mixed etiologies; for example, the Toronto HCC risk index (THRI), composed of age, sex, etiology, and platelet count, was externally validated in a cirrhosis cohort [18]. The aMAP (age, male, albumin–bilirubin, and platelets) score, calculated by using age, sex, albumin, total bilirubin, and platelet count, was derived from a cohort of HBV-infected patients and independently validated in nine cohorts with various etiologies [19]. In a large VA patient population, HCC-risk-predictive algorithms were developed according to specific clinical contexts, i.e., HCV infection (pre- and post-antiviral treatment), alcoholic cirrhosis, and NAFLD cirrhosis, and they were implemented in a publicly available web application, HCC risk calculator (hccrisk.com) [20,21,22]. Prognostic performance of the models, measured by concordance index, ranged approximately from 0.60 to 0.80 across various clinical contexts. Among the utilized laboratory tests, AFP is frequently incorporated in several HCC risk scores, especially in the context of therapeutically cured HCV infection (Table 1). This may be because HCV cure diminishes AFP elevation due to non-specific hepatic inflammation caused by active viral replication [23]. Studies have suggested that specific clinical contexts defined by several factors likely affect accuracy of HCC risk prediction. For example, the ever-evolving antiviral therapies for HBV and HCV will substantially alter the baseline HCC risk level depending on the status of viral control [19,24,25]. In addition, as suggested from multiple studies on association of germline genotypes with HCC risk, it is plausible that some of such HCC-risk-associated genotypes are bound to patient race/ethnicity and may guide tailored application of HCC risk prediction by geographic representation of racial/ethnic background. Furthermore, dietary habits and/or food contamination with carcinogens such as aflatoxin could be linked to certain geographic regions, possibly allowing a region-tailored strategy of HCC risk assessment [26]. Proper consideration for these factors and incorporation in the risk-prediction algorithm may improve accuracy of HCC risk prediction and enhance cost-effectiveness of HCC screening tailored by these parameters. While most clinical-variable-based scores were derived based on regression models, a sophisticated machine learning approach has also been utilized. Ioannou et al. identified a recurrent neural network-based clinical score to predict incident HCC within three years in 48,151 HCV-infected patients with cirrhosis and showed the new score outperformed conventional linear-regression-model-based score [27]. Despite the promise of the recently emerging utilization of machine learning and artificial intelligence approach to develop risk-predictive models, several limitations are worth noting. First, methodologies such as multi-layer neural network are prone to overfit data structure of specific training dataset, which may diminish generalizability of the model [28]. In addition, it is possible that the modeling disregard certain patient subgroups depending on the structure of the training set. To mitigate these concerns, it will be increasingly important to ensure transparency of the modeling process and clarification of potential pitfalls [29,30].

Semi-quantitative histological fibrosis stage has been reported to be associated with HCC risk, although sampling variability should be concerned when evaluating the liver biopsy specimen [31]. Quantification of collagen proportionate area in liver biopsy specimen is more quantitative measure of liver fibrosis, which is associated with the risk of hepatocarcinogenesis [32,33,34]. Deep-learning algorithm applied to histological images may have the potential to improve the histopathological morphology-based approach of HCC risk assessment [35,36]. Saillard et al. identified histological features predicting survival after surgical resection, supporting feasibility to extend this approach to predict de novo HCC risk [36]. Liver stiffness measurement (LSM) by ultrasound- or MRI-based elastography has been associated with elevated risk of HCC, especially in the settings of viral hepatitis and cured HCV infection [37,38,39,40]. López et al. recently showed that change in LSM one year after achieving HCV cure, as well as baseline LSM, was independently associated with long-term HCC risk in patients with compensated advanced liver fibrosis [41]. In addition, predictive performance of REACH-B score, a clinical score for HBV-associated HCC risk, can be improved by incorporating LSM into the model [42,43].

## 3. Molecular HCC Risk Biomarkers

Molecular information has a potential to improve the accuracy of HCC risk prediction, in combination with the clinical HCC risk scores (Table 2). Molecular risk biomarkers can be integrated with clinical variables/scores as composite scores, or serially assessed with clinical risk scores for step-wise enrichment of high-risk patient population. In addition, HCC risk biomarkers may be used as selection markers to identify a subset of patients who will benefit from HCC-preventive intervention. They may also serve as surrogate endpoints in HCC prevention clinical trials when their therapeutic modulation reflects change in future HCC risk.

### 3.1. Germline DNA Variants

Single nucleotide polymorphism (SNP) is a major type of germline DNA polymorphism with a wide variety of pathogenic implications. Previous genome-wide association studies (GWASs) have revealed several SNPs that are likely associated with genetic susceptibility to HCC [65] (Table 2). Germline SNPs can be easily assessed by using a buccal swab or peripheral blood sample, at any time point, because they do not change throughout life and are increasingly more accessible with decreasing costs over time as a viable tool for potential molecular HCC risk prediction. The association of these SNPs with HCC risk is generally modest (odds ratios (ORs) of ~1.5 or less). However, a combination of multiple SNPs may achieve improved performance in HCC risk estimation [66]. Of note, some SNPs seem to be associated with liver disease etiology, suggesting etiology-dependent mechanisms of hepatocarcinogenesis [67].

Several SNPs are reported to be associated with HCV-related HCC. A SNP of *IFNL3* (also known as *IL28B*, encoding one of the cytokines) is associated with risk of HCC development, especially in patients with or without sustained virologic response (SVR) of HCV infection [68,69,70,71,72]. The SNP was initially identified as a predictor of spontaneous clearance of HCV against interferon-based antiviral therapy, and later was found to be related with *IFNL3*-*IFNL4* haplotype-dependent hepatic inflammation and fibrosis [73,74,75,76,77]. A SNP in *MICA* gene, an immune-related gene encoding the highly polymorphic major histocompatibility complex class I chain-related protein A, was shown to be related to HCC risk in Japanese HCV-infected patients (adjusted OR, 1.36) [78]. This result was validated in an Asian cohort, whereas opposite results were reported in a Swiss cohort, suggesting that the HCC risk association may be patient-race/ethnicity dependent [79,80,81]. A SNP in *DEPDC5* gene, encoding a protein that inhibits the mTORC1 pathway, was identified as associated with HCV-related HCC (adjusted OR, 1.96) in Japanese patients [82]. This SNP was also associated with progression of liver fibrosis in European patients [83]. A SNP in the intron of *TLL1* gene, encoding a type of matrix metalloprotease that has relation to the liver development, was associated with HCC risk after SVR by interferon-based antiviral therapy in a Japanese cohort (adjusted hazard ratio (HR), 1.78), whereas this SNP was not associated with HCC risk after SVR by direct-acting antiviral (DAA) in a Caucasian cohort [84,85]. *EGF* 61*G allele was associated with HCC risk in Eastern and Western patients with HCV or HBV infection (pooled OR, 1.38) [86,87,88].

HCC-risk-associated SNPs have also been explored in HBV-related HCC. A SNP in *KIF1B* gene, encoding one of the motor proteins, was first reported to be associated with HCC risk in Chinese HBV-infected patients and validated in five independent Chinese cohorts, whereas the reports from Korea, Japan, and Thailand failed to validate the result [89,90,91]. Another GWAS study from China identified SNPs in *STAT4* and *HLA-DQB1*/*HLA-DBA2* (adjusted OR, 1.21 and 1.49, respectively), both of which were associated with progression of liver fibrosis [92,93]. Genomic DNA duplication at chromosome 15q13.3 was identified as a high-risk variant (OR, 12.02) by a germline copy number variation-based GWAS of Chinese HBV carriers, though its prevalence is low (2.3% and 0.2% in HCC cases and controls, respectively) [94].

One of the major SNPs associated with metabolic HCC is *PNPLA3* I148M valiant, initially identified as a risk variant for presence of NAFLD [95]. The protein encoded by *PNPLA3* is located on lipid droplet in hepatocytes and hepatic stellate cells and elicits hydrolase activity on triglycerides and retinyl esters, respectively [96]. The *PNPLA3* I148M variant causes impaired triglyceride mobilization and accumulation of lipid droplet by evading ubiquitylation and by comparative gene identification–58 (CGI–58)–dependent inhibition of adipose triglyceride lipase, resulting in hepatic steatosis [97,98,99]. *PNPLA3* rs738409 G allele was associated with elevated HCC risk in NAFLD and alcohol-related liver disease patients [67,100]. The *PNPLA3* I148M was also associated with a high liver-related mortality in a large population study and liver cancer-related mortality in patients with NAFLD in the US [101,102] Further, this variant may be associated with HCC risk after achieving SVR with DAA, [103,104] presumably reflecting the association between HCV infection and hepatic steatosis [105].

A SNP in *TM6SF2*, encoding E167K substitution, was originally identified as one of the NAFLD-associated SNPs [106]. Subsequent studies have revealed its association with the risk of alcoholic cirrhosis and alterations of hepatic steatosis in viral hepatitis [107,108]. The protein encoded by *TM6SF2*, transmembrane 6 super family member 2A, is a regulator of liver lipid metabolism, and knockdown/knockout of *Tm6sf2* in mice caused an increase of hepatic triglyceride content [106,109,110]. *TM6SF2* E167K was associated with HCC in alcohol-related cirrhosis [111]. Homozygous minor allele of the locus was associated with an increased risk of NAFLD-HCC as well, although the significance was lost after adjusting for clinical confounding factors [112].

*MBOAT7*-*TMC4* variant rs641738 was initially reported as a risk SNP for alcoholic cirrhosis together with *TM6SF2* E167K [106]. This variant also showed an association with the development and severity of NAFLD in Caucasian patients (adjusted OR, 1.30) and with NAFLD-related HCC even in patients without advanced liver fibrosis (adjusted OR, 2.10) [113,114]. *MBOAT7* deletion was reported to cause fatty liver by activation of SREBP-1c [115]. A SNP in *DYSF* gene was associated with an elevated risk of NASH-related HCC in Japanese patients [116]. Short telomeres and germline mutations in *TERT* gene were observed in NAFLD-related Caucasian HCC patients [117]. A splice variant in *HSD17B13* was negatively associated with histological NASH severity, progression to cirrhosis, and HCC development [118,119,120].

The combination of multiple SNPs has also been thought to improve the ability to predict the individual HCC risk. Gellert-Kristensen et al. combined three NAFLD-related SNPs in *PNPLA3*, *TM6SF2*, and *HSD17B13* and showed the association with HCC by comparing with the general population (OR, 29.0 for the highest risk group compared to the lowest risk group), although application in the general population is unlikely [66]. Degasperi et al. evaluated another combination, consisting of SNPs in *PNPLA3*, *MBOAT7*, *TM6SF2*, and *GCKR*, that was associated with liver fat content in cirrhosis patients after curing HCV with DAA (adjusted HR, 2.30), although the cutoff to define the high-risk group was not independently validated [103].

### 3.2. Hepatic Transcriptomic Signatures

Transcriptomic changes in diseased liver tissue have been widely studied as a direct and reliable measure of pathogenic molecular dysregulation associated with HCC risk and liver disease progression [158]. Hepatic tissue transcriptome profiling has been widely used as a reliable resource to explore biological dysregulation associated with patient prognosis and clinical phenotypes. Earlier studies analyzed diseased liver tissues obtained at the time of hepatic surgery performed as treatment of liver tumor [14]. These studies demonstrated the proof of concept that hepatic transcriptomic dysregulation is associated with the risk of developing de novo HCC after the surgical therapies [159]. However, the HCC-risk-predictive performance should be re-evaluated in HCC-naïve patients, for the prediction of future HCC risk. In addition, the benefits of performing liver biopsy should be justified over the potential risk of adverse events caused by the procedure, such as bleeding. Thus, surrogate markers of hepatic transcriptome will overcome the limitation and enable clinically applicable HCC risk prediction with enhanced accuracy based on molecular information. Prognostic liver signature (PLS) is a hepatic transcriptome-based signature including 186 genes, initially derived from resected liver tissues adjacent to curatively treated HCC and was predictive of overall survival and late HCC recurrence after the treatment [159]. Subsequent studies showed that the PLS and its reduced version (32-gene signature) can predict de novo incidents of HCC in HCV-related early stage cirrhosis, as well as de novo HCC recurrence after curative HCC resection in patients with all major HCC etiologies, including HBV, HCV, alcohol abuse, or NAFLD (HRs, 2.65-10.94) [133,134,135]. In addition, PLS enabled quantitative HCC risk estimation even in chronic hepatitis C patients after achieving SVR [134,160]. Of note, the PLS was implemented in an FDA-approved clinical diagnostic assay platform [160]. A hepatic injury and regeneration (HIR) signature was derived from HCC-surrounding liver tissues from HBV-related HCC patients and was validated for late HCC recurrence (adjusted HR, 2.2) in Asian patients [136].

Transcriptome signatures associated with specific cell types and histological features have also been proposed. Hepatic stellate cells (HSCs) are the liver-specific mesenchymal cells that reside in the space of Disse and are the major precursor of myofibroblasts that drive liver fibrogenesis and create carcinogenesis-promoting hepatic tissue microenvironments [161]. Two HSC-associated signatures have been reported with validation in human cohorts [133,134]. A gene expression signature of 12 chemokine genes was reported to be associated with ectopic lymphoid structure (ELS), a tertiary lymphoid structure composed of lymphocytes and dendritic cells [162]. The ELS signature was associated with late HCC recurrence in a human cohort (adjusted HR, 3.58), suggesting its association with de novo HCC in chronically inflamed liver [139]. A 172-gene signature, named immune-mediated cancer field (ICF), was associated with deregulated immune response in liver and the risk of HCC development [140].

### 3.3. Somatic DNA Mutations in Non-Malignant Liver

Somatic DNA mutations are, in general, implicated as oncogenic drivers in a variety of cancer types, including HCC. Recent studies have suggested that more somatic DNA mutations also accumulate in cirrhotic livers, as compared to healthy livers, and possibly affect propensity of carcinogenic; moreover, they could be detected in liver tissues and in circulation [65,163,164,165]. In addition, a recent study suggested that somatic mutations in *PKD1*, *KMT2D*, and *ARID1A* genes in non-malignant cirrhotic liver protect hepatocytes from malignant transformation [157]. It is noteworthy that the same somatic events, such as the dysregulation of the *ARID1A* gene, can either promote or suppress HCC initiation and progression depending on the stage of hepatocarcinogenesis. In addition, rodent-model-based studies have suggested that hepatocyte polyploidy confers protection from HCC development, which can be therapeutically induced by modulation of a cytokinesis gene/protein, anillin [166]. To be considered as biomarkers for HCC risk prediction, detectability in clinically accessible specimens will be a key issue for this type of molecular information.

### 3.4. Circulating Biomolecules

AFP has been reported as a biomarker for not only an early HCC detection but a risk stratification. Hughes et al. revealed that a longitudinal change in AFP can more accurately capture future HCC risk [23]. Further, biological-hypothesis-driven studies have identified several circulating protein biomarkers for HCC risk, including serum insulin-like growth factor I (IGF-I), osteopontin (OPN), and interleukin-6 (IL-6) [152,153,167]. In addition, various types of biomolecules released into body fluids have been suggested as HCC-related biomarkers [14]. In a nested case–control study from a prospective EPIC cohort, including 129 cases and 1:1 matched controls (median follow-up, 6.2 years), untargeted liquid chromatography–mass spectrometry (MS)-based metabolomics identified 14 metabolites associated with long-term HCC risk, including nine high-risk (N1-acetylspermidine, isatin, p-hydroxyphenyllactic acid, tyrosine, sphingosine, L,L-cyclo(leucylprolyl), glycochenodeoxycholic acid, glycocholic acid, and 7-methylguanine) and five low-risk (retinol, dehydroepiandrosterone sulfate, glycerophosphocholine, γ-carboxyethyl hydroxychroman, and creatine) metabolites (ORs for high-risk and low-risk metabolites, 2.16–6.78 and 0.27–0.56, respectively, for incremental one standard deviation) [155].

Cell-free (cf) nucleic acids in circulation have been explored as sources to detect pathogenic alterations, such as DNA mutations, epigenetic modification, and aberrant abundance of miRNA in an organ of interest. These altered molecules likely reflect molecular dysregulations in subclinical pre/neoplastic cells before clinical detection of tumor, and therefore such circulating nucleic acid-based biomarkers will serve for early tumor detection and/or short-term HCC risk prediction. For instance, a recent study from four Chinese centers introduced an assay detecting point mutations in *TP53*, *CTNNB1*, *AXIN1*, and *TERT* promoter, as well as HBV integration in cfDNA for early detection of HBV-related HCC, which was combined with AFP, des-γ-carboxy prothrombin, and relevant clinical variables, to develop the “HCCscreen” algorithm [141]. In a validation cohort of 331 AFP- and ultrasound-negative individuals, the algorithm identified 24 test positive patients, among whom four were diagnosed with HCC within six to eight months (17% positive predictive value) [157].

## 4. Conclusions and Future Perspectives

Clinical translation and implementation of HCC risk scores and biomarkers should be based on proper evidence of validation of their risk-predictive performance as outlined in guidance/recommendation in general oncology and HCC care, because a retrospective assessment intrinsically contains biases, such as unmeasured confounders and selection biases [13,168,169]. However, for risk-predictive biomarkers, their prospective evaluation has been the major bottleneck due to the long-term period required for clinical follow-up to observe a sufficient number of clinical outcomes of interest for statistically detectable prognostic association [170]. Such long-term prospective clinical follow-up is costly, and the likelihood of successful validation is generally low. To mitigate the challenge, an alternative strategy has been proposed. Prospective-specimen-collected design, retrospective-blinded-evaluation (PRoBE) design, or prospective–retrospective design aims to prospectively collect biospecimens, with an intention to test unspecified biomarkers, which are subsequently utilized in a retrospective manner, leveraging matured clinical follow-up data [13,64]. Before clinical application of the clinical risk scores and biomarkers, their performance should be fully ascertained on two aspects. First, when a score/biomarker relies on certain devices such as imaging modality or biochemical test, their technical validity should be assessed for reproducibility of the measurements across test dates and sites based on certain technical implementation that will be used in an actual clinical setting. Second, the magnitude of risk association should be assessed in independent patient series, with in-depth consideration on patient demographics and characteristics, to evaluate their effects on the performance. In assessing the real-world cost-effectiveness of a risk-predictive approach, it is important to consider patient/physician acceptability and adherence, particularly because it is known that the HCC screening utilization rate is low due to a variety of patient- and practitioner-related reasons [11]. This can be estimated by incorporating the HCC screening utilization rate into the simulation model, as performed in a recent Markov-model-based cost-effectiveness analysis [12]. In addition, such a modeling approach informs the desired performance of risk score or biomarker to enable cost-effective HCC screening. For generic HCC biomarker evaluation, several resources have been developed, including the Hepatocellular Carcinoma Early Detection Strategy (HEDS) study [171] and Texas Hepatocellular Carcinoma Consortium (THCCC) [172]. These prospective cohorts will accelerate the validation of risk-predictive biomarkers and facilitate their clinical translation upon successful validation. These expanding resources following the PRoBE design are expected to provide more reliable measures of effect size in risk estimation, which will inform the design and required sample size in subsequent interventional studies to examine magnitude of clinical benefit from the risk-stratified approach. Such evidence will eventually support the decision of whether or not the tailored strategy can be incorporated into clinical practice. In conclusion, several promising development of HCC risk scores and biomarkers are underway, and they are expected to transform the “one-size-fits-all” strategy and contribute to the substantial improvement of the poor prognosis of HCC patients in the foreseeable future.

## Figures and Tables

**Figure 1 jcm-09-03843-f001:**
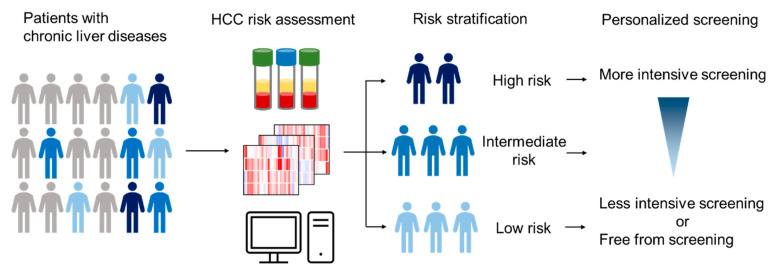
Stratified hepatocellular carcinoma (HCC) screening tailored by predicted individual HCC risk.

**Table 1 jcm-09-03843-t001:** Clinical HCC risk scores.

Risk Score	Variables	Study Design	Registry *	Endpoint (HCC)	Major Etiology	Region/Country	Race	Cirrhosis **	Validation ^☨^	Reference
UM regression model	Machine-learning (23 clinical variables)	Cohort	Prospective–retrospective	Development	HCV, cryptogenic, alcohol, other	USA	Caucasian, Black, Hispanic	100% + 41%	External	[44]
aMAP risk score	Age, sex, albumin–bilirubin, platelets	Cohort	Prospective–retrospective	Development	HBV, HCV, HCV after SVR, non-viral	International	Asian, Caucasian	19.3% + 11.4–100%	External	[19]
ADRESS-HCC	Age, diabetes, race, etiology, sex, Child-Pugh score	Cohort	Retrospective	Development	HCV, alcohol, NASH, HBV, other	USA	Non-Hispanic white, Hispanic/Latino, African American, Asian	100% + 100%, 100%	External	[45]
THRI	Age, sex, etiology, platelets	Cohort	Retrospective	Development	Viral, steatohepatitis, PBC, AIH	Canada	n.r.	100% + 100%	External	[18]
Hughes et al.	AFP	Cohort	Prospective–retrospective	Development	HCV, HBV	Japan, Scotland	n.r.	n.r.	External	[23]
CU-HCC	Age, albumin, bilirubin, HBV-DNA, cirrhosis	Cohort	Prospective–retrospective	Development	HBV	Hong Kong	n.r.	38%	External	[46]
LSM-HCC	Liver stiffness, age, albumin, HBV-DNA	Cohort	Prospective	Development	HBV	Hong Kong	n.r.	31%	External ^‡^	[47,48,49]
REACH-B	Sex, age, ALT, HBeAg, HBV-DNA	Cohort	Prospective–retrospective	Development	HBV	Asia	n.r.	0% + 18%	External	[42]
NGM1-HCC	Sex, age, family history of HCC, alcohol, ALT, HBeAg	Cohort	Prospective–retrospective	Development	HBV	Taiwan	n.r.	n.r.	External ^‡^	[50,51]
NGM2-HCC	Sex, age, family history of HCC, alcohol, ALT, HBV-DNA	Cohort	Prospective–retrospective	Development	HBV	Taiwan	n.r.	n.r.	External ^‡^	[50,51]
GAG-HCC	Age, sex, HBV-DNA, core promoter mutations, cirrhosis	Cohort	Retrospective	Development	HBV	Taiwan	n.r.	15%	External ^‡^	[35,40,41]
FIB-4	FIB-4 (AST, ALT, platelets, age)	Cohort	Retrospective	Development	HBV	S. Korea	n.r.	10%	External ^‡^	[52,53]
PAGE-B	Age, sex, platelets	Cohort	Retrospective	Development	HBV treated with NA	Europe	Caucasian	20% + 48% *	External	[25]
D2AS risk score	HBV-DNA, sex, age	Cohort	Retrospective	Development	HBV	S. Korea	Asian	0%	External	[54]
CAMPAS model score	Cirrhosis, age, sex, platelets, albumin, liver stiffness	Cohort	Retrospective	Development	HBV treated with NA	S. Korea	Asian	40%	External	[55]
AASL-HCC score	Age, albumin, sex, cirrhosis	Cohort	Retrospective	Development	HBV treated with NA	S. Korea	Asian	39% + 39%	External	[56]
modified PAGE-B	Age, sex, platelets, albumin	Cohort	Retrospective	Development	HBV treated with NA	S. Korea	Asian	19% + 20%	External ^‡^	[57,58]
CAMD score	Cirrhosis, age, sex, diabetes mellitus	Cohort	Retrospective	Development	HBV treated with NA	Taiwan, Hong Kong	Asian	26% + 7%	External	[59]
Ganne-Carri et al.	Age, alcohol, platelets, GGT, SVR	Cohort	Prospective–retrospective	Development	HCV	France	n.r.	100%	External ^‡^	[60,61]
REVEAL-HCV	Age, ALT, AST/ALT ratio, HCV-RNA, cirrhosis, HCV genotype	Cohort	Prospective–retrospective	Development	HCV	Taiwan	n.r.	1% + 7%	External	[62]
ADRES score	SVR24, sex, FIB-4 index, AFP	Cohort	Retrospective	Development	HCV-SVR treated with DAA	Japan	Asian	n.r.	External	[24]
Sinn et al.	Age, sex, smoking, diabetes, total cholesterol, ALT	Cohort	Retrospective	Development	non-HCV, HBV, alcohol	S. Korea	Asian	general population	External	[63]

* “Prospective–retrospective” indicates a retrospective analysis of prospectively collected cohort in the past [64]. ** Training + validation. ^☨^ Validation: “Internal”, validation in patients from the same institution(s); “External”, validation in patients from independent institution(s). ^‡^ Validated by subsequent study. AFP, alpha-fetoprotein; AIH, autoimmune hepatitis; ALP, alkaline phosphatase; ALT, alanine aminotransferase; AST, aspartate aminotransferase; DAA, direct-acting antiviral agent; GGT, gamma-glutamyltransferase; HbA1c, glycosylated hemoglobin, type A1c; HBV, hepatitis B virus; HBeAg, hepatitis B e antigen; HCV, hepatitis C virus; HCC, hepatocellular carcinoma; NA, nucleoside/nucleotide analogues; NASH, non-alcoholic steatohepatitis; n.r., not reported; PBC, primary biliary cholangitis; SVR, sustained virologic response; UM regression model, University of Michigan regression model; aMAP, age, male, albumin–bilirubin, and platelets risk score; ADRESS-HCC, age, diabetes, race, etiology of cirrhosis, sex, and severity of liver dysfunction HCC; THRI, Toronto hepatocellular carcinoma risk index; CU-HCC, Chinese University-HCC; LSM-HCC, liver stiffness measurement for HCC; REACH-B, risk estimation for hepatocellular carcinoma in chronic hepatitis B; NGM1 or 2-HCC, nomogram 1 or 2-HCC; GAG-HCC, guide with age, gender, HBV-DNA, core promoter mutations and cirrhosis-HCC; FIB-4, fibrosis-4; PAGE-B, platelets, age, gender-HBV; CAMPAS model score, cirrhosis on ultrasonography, age, male gender, platelet count, albumin and liver stiffness; AASL-HCC score, age, albumin, sex, liver cirrhosis HCC score; CAMD score, cirrhosis, age, male sex, and diabetes mellitus score; REVEAL-HCV, risk evaluation of viral load elevation and associated liver disease/cancer in HCV; ADRES score, after DAAs recommendation for surveillance score.

**Table 2 jcm-09-03843-t002:** Molecular HCC risk biomarkers.

Type of Biomarker	Biomarkers/Scores	Variables	Study Design	Assessment *, **	Endpoint (HCC)	Major Etiology	Region/Country	Race	Cirrhosis ^☨,^^‡^	Combined Clinical Variables	Validation ^§^	Reference
SNP												
	*EGF*	*EGF 61AG* (rs4444903, A>G)	Case–control (Meta-analysis)	Retrospective	Presence	HBV, HCV	France, Italy, China, Egypt, Japan, USA	Asian, European, African	n.r.	n.a.	External	[88]
	*IFNL3*	*IFNL3* (rs12979860: C>T, rs8099917: T>G)	Case–control, cohort (Meta-analysis)	Retrospective	Presence	HCV, HBV	Multiple (Asia, Europe)	n.r.	n.r.	n.a.	External ^☨☨^	[68,121]
	MICA	*MICA* (rs2596542, C>T)	Case–control (Meta-analysis)	Retrospective	Presence	HCV, HBV	Egypt, China, Japan, Vietnam, Italy, Switzerland	Asian, European	14%	n.a.	External	[122]
	*DEPDC5*	*DEPDC5* (rs1012068: T>G)	Case–control	Retrospective	Presence	HCV	Japan	Asian	n.r.	n.a.	No	[123]
	*TLL1*	*TLL1* (rs17047200: A>T)	Case–control	Retrospective	Presence	HCV after SVR treated with IFN	Japan	Asian	25% + 20% (F3-4)	Age, albumin, AFP after SVR	External ^☨☨^	[84,124]
	*KIF1B* or 1p36.22	*KIF1B* or 1p36.22 (rs17401966, A>G)	Case–control (Meta-analysis)	Retrospective	Presence	HBV	China, Japan, S. Korea, Thailand	Asian	n.r.	n.a.	External	[125]
	*STAT4*	*STAT4* (rs7574865, G>T)	Case–control (Meta-analysis)	Retrospective	Presence	HBV	Thailand, China, Vietnam, S. Korea	Asian	n.r.	n.a.	External	[126]
	*HLA-DQB1/HLA-DBA2*	*HLA-DQB1/HLA-DBA2* (rs9275319 A>G)	Case–control	Retrospective	Presence	HBV	China	Asian	n.r.	n.a.	External ^☨☨^	[92,127]
	*PNPLA3*	*PNPLA3* (rs738409: C>G)	Case–control, cohort (Meta-analysis)	Retrospective	Presence	NAFLD, alcohol, HCV	Europe, Japan	Caucasian	n.r.	n.a.	External	[128]
	*TM6SF2*	*TM6SF2* (rs58542926: C>T)	Case–control (Meta-analysis)	Retrospective	Presence	Alcohol	Italy, Thailand, France, Germany	Caucasian	n.r.	n.a.	External	[129]
	*MBOAT7*	*MBOAT7* (rs641738: C>T)	Case–control	Retrospective	Presence	NAFLD	Italy	Caucasian	28 % (F3–4)	n.a.	No	[114]
	*HSD17B13*	*HSD17B13* (rs72613567: TA)	Case–control	Retrospective	Presence	Alcohol	Germany	Caucasian	100%	n.a.	External ^☨☨^	[130,131,132]
Score of SNPs												
	Genetic risk score	SNPs of *PNPLA3, TM6SF2, HSD17B13*	Cohort	Prospective-retrospective	Development	General population	Denmark, UK	Caucasian + n.r.	0.4% + 0.1%	Alcohol cirrhosis, ALT	External	[66]
	Fat-genetic risk score (hepatic fat genetic risk score)	SNPs of *PNPLA3, TM6SF2, MBOAT7, GCKR*, and hepatic fat content	Cohort	Prospective-retrospective	Development, recurrence	HCV treated with DAA	Italy	Italian, Egyptian	100%	Sex, diabetes, albumin	No	[103]
Tissue transcriptome												
	Prognostic liver signature (PLS)	186-gene signature	Cohort	Prospective-retrospective	Development, recurrence	HCV	Training Italy, Validation USA	Asian, Caucasian	100%	AFP, vascular invasion, bilirubin, platelet, Child–Pugh class, AJCC stage	External ^☨☨^	[133,134,135]
	HIR gene signature	233/65-gene signature	Cohort	Retrospective	Late/early recurrence	HBV	S. Korea, Hong Kong, China	Asian	53% + 63% + 93%	n.a.	External	[136]
	Activated HSC gene signature	37-gene signature	Cohort	Retrospective	Recurrence	HBV	China	Asian	91%	Child-Pugh staging	External	[137]
	HSC signature	122-gene signature	Cohort	Prospective-retrospective, retrospective	Development	HCV, HBV	USA	Caucasian, Asian	100%	Bilirubin, platelets	Internal	[138]
	Ectopic lymphoid structure signature	12-gene signature	Cohort	Retrospective	Late recurrence	HCV	Germany	Asian	52%	n.a.	No	[139]
	Immune mediated cancer field signature	172-gene signature	Cohort	Retrospective	Development	HCV	International	Caucasian	n.r.	Bilirubin	No	[140]
Circulating												
	cfDNA	mutations of 4 genes, HBV integration	Cohort	Retrospective	Development	HBV	China	Asian	11%	AFP, US	Internal	[141]
	miRNA	7/8 miRNAs	Cohort	Prospective	Development	HBV	USA	Asian	35%	AFP	No	[142]
	miRNA	5 miRNAs	Cohort	Prospective-retrospective	Development	HBV, HCV	Taiwan	Asian	100%	HCV	No	[143]
	DNA methylation	*TBX2* hypermethylation	Nested case–control	PRoBE	Development	HBV, HCV, alcohol	Taiwan	Asian	n.r.	n.a.	No	[144]
	GlycoHCCRiskScore	serum protein N-glycans	Case–control	Prospective-retrospective	Development	HCV	Belgium	Caucasian	100%	n.a.	No	[145]
	Serum glycan	M2BPGi	Cohort	Retrospective	Development	HCV	Japan	Asian	17%	F4, AFP, age, response to IFN therapy	External ^☨☨^	[146,147,148,149]
	Cytokine	IL-6	Cohort	Prospective-retrospective	Development	HCV	Japan	Asian	n.r.	Sex, age, platelets (female), AFP, prothrombin time activity (male), alcohol, BMI (female)	External ^☨☨^	[150,151,152]
	Protein	IGF-1	Cohort	Prospective	Development	HCV	Italy	Caucasian	100%	n.a.	No	[153]
	HCC risk score	2 amino acids (Phe, Gln)	Cohort	Retrospective	Development	HBV, HCV	Taiwan	Asian	n.r.	age, HCV	No	[154]
	Metabolites	14 metabolites	Nested case–control	Prospective-retrospective	Development	HBV, HCV, Alcohol	Europe	n.r.	n.r.	n.a.	No	[155]
	Metabolites	2 metabolites (phenylalanyl-tryptophan, glycocholate)	Nested case–control	PRoBE	Development	HBV	China	Asian	n.r.	AFP	No	[156]

* “Prospective-retrospective” indicates a retrospective analysis of prospectively collected cohort in the past [64]. ** PRoBE, prospective specimen collection, retrospective blinded evaluation study design [157]. ^☨^ Case: control. ^‡^ Training + validation. ^§^ Validation: “Internal”, validation in patients from the same institution(s); “External”, validation in patients from independent institution(s). ^☨☨^ Validated by subsequent study. AFP, alpha-fetoprotein; AJCC, American Joint Committee on Cancer; cfDNA, circulating free DNA; DAA, direct-acting antiviral agent; HBV, hepatitis B virus; HCV, hepatitis C virus; HCC, hepatocellular carcinoma; HIR, hepatic injury and regeneration gene expression; HSC, hepatic stellate cell; IFN, interferon; M2BPGi, mac-2 binding protein glycosylation isomer; miRNA, microRNA; n.a., not applicable; NA, nucleoside/nucleotide analogue; NAFLD, non-alcoholic fatty liver disease; NASH, non-alcoholic steatohepatitis; n.r., not reported; SNP, single-nucleotide polymorphism; SVR, sustained virologic response; US, ultrasound.

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
