# Peer review of "Clinical and Molecular Prediction of Hepatocellular Carcinoma Risk"

_jcm, 2020, doi:10.3390/jcm9123843_

Round 1
Reviewer 1 Report
This is an important topic and a generally well written manuscript. As the authors highlight hepatocellular carcinoma (HCC) surveillance is an area of debatable cost efficiency and improving enrollment and tailoring patient specific surveillance has the opportunity to optimise cost efficiency and outcomes. The authors have taken a large topic area and aimed describe important studies in the field. It is not clear however how these studies were identified; a systemic approach would be optimal.
Overall I enjoyed the review and particularly found tables 1/2 a useful resource and reference. They are however incomplete and this would be aided by a systematic approach. I would highlight a couple of examples where studies are not included in the table which would suggest they have not been detected by the approach taken by the authors. For example personally I use the HCC risk score available at HCCrisk.com which is formed from these two papers both I would recommend including in table 1. DOI: 10.1016/j.jhep.2018.07.024, DOI: 10.1016/j.jhep.2019.05.008. I would also suggest adding the study by Poynard et al. doi: 10.1111/apt.15082 to the list in Table 1. Clarification of the approach taken by the authors for this work should be provided.
Generally, the text is a somewhat superficial overview of the field and would benefit from expanding in some areas (examples given below). In general the discussion of the topic is rather descriptive and doesn’t not give any general advice on relative benefits/pitfalls of the various scores. Exhaustive discussion and clinical recommendations may be beyond the scope of this article however.
Major comments
With conditions that underlie HCC risk continually evolving (i.e. advent of mass Hepatitis B vaccination programmes, effective Hepatitis C therapies and the epidemiological shift toward NAFLD), it is likely that the multiple HCC risk prediction models will evolve in terms of their calibration. Similarly, many of these were developed in specific populations (i.e. east Asian or western Europe or the Americas) where genetic determinants and underlying risk factors for HCC are likely to be substantially different. This is not addressed and is important to ensure equitable and effective surveillance for at risk groups.
There is mention of machine learning throughout, but no real insight provided into the methods used, how the models were validated and the development process used. It is important these are addressed, as there are known issues with machine learning / artificial intelligence producing models that discriminate against certain population groups (i.e. women/ ethnicities). How can clinical scores that use these approaches become more ethical and equitable?
The conclusion proposes that HCC risk scores/ biomarkers should be based on full evaluation and performance. However, the review does not touch on the quality of evaluation or the biases contained in many of these studies. Similarly, there is no mention of perhaps the most important aspects to evaluate in any population-based approach – cost and patient acceptability. Some additional discussion around what sorts of testing would be clinically cost effective and acceptable for patients would be beneficial. I would also suggest the addition of potential alternative strategies for screening based upon stratification. Such a use of cheaper/more expensive methods for stratified patients rather than just altering the frequency. This is, as far as I am aware, an evidence free zone, but conceivably there may be a role for more targeting surveillance with for example cfDNA, GALAD, the use of dynamic biomarker analysis in high risk patients (e.g. DOI: 10.1111/biom.12717 and doi: 10.1371/journal.pone.0156801), or potentially taking patients at high risk clinical score and then using transcriptomics to further refine their risk scores. This area could be discussed. The trials for altering screening frequency from 3-12 months were in an unstratified population but pragmatically six months makes sense in relation to tumor doubling time; see http://dx.doi.org/10.1136/gutjnl-2020-321040. Therefore, I would caution the suggestion for less frequent surveillance as this may just lead to late stage detection and negate any benefit of performing surveillance at all. In support of this there is some emerging data to suggest that poor adherence to surveillance is associated with equivalent outcomes to no surveillance at all. Similarly, I would be interested in the authors’ opinion on what would be deemed sufficient from any of these risk scores to remove a patient from ongoing surveillance completely. I would however discourage the provision of clinical advice within this article and leave this area to the guidelines of the major associations.
As the annual risk of developing HCC is relatively low, the limiting factor for uptake of HCC surveillance in practice is cost (to either healthcare system or patients). With the burdens of disease in low-middle incomes shifting to non-communicable diseases, for example, how could these be made applicable for future generations of patients at risk of HCC?
In the transcriptome section, the study flits between discussing methods of early detection in patients at risk of HCC (i.e. populations with no HCC, but with preexisting liver disease) and those who have had HCC and markers of post surgical recurrence. These are two very different groups and two very different clinical scenarios. For example, at the population level obtaining liver biopsies on any patient at risk of HCC is not feasible and posses additional risk to patients. Whilst performing transcriptomic on resection tissue is relatively simpler but still requires specific clinical pathways and has cost/resource implications.
Similarly embarking upon transcriptomic signatures is more expensive and reliant on liver tissue (as highlighted above). The relevance between resection based transcriptome and a smaller biopsy sample should be discussed. Similarly the risk of performing a biopsy in a patient purely to obtain a cancer risk transcriptome should be weighed against the risk of the biopsy itself if not being performed for other clinical indications. I also feel that a discussion regarding the now appreciated clonality of cirrhotic nodules and the present of cancer associated mutations in the tissue are relevant here doi: 10.1016/j.cell.2019.03.026 and doi: 10.1038/s41586-019-1670-9. Relevant review (e.g. doi: 10.1016/j.jhep.2020.01.019).
The roles of mutational signatures or mutational load in liver tissue is not discussed, just bulk transcriptomics. These mutational signatures could be used to assist in defining aetiology as well as HCC risk. This could be included and discussed briefly but is understandably outwith the scope of clinical practice currently.
I would also be explicit for the general reader that because the mutations being discussed in section 3.1 are germline this could be tested once to inform lifetime risk. The costs of assaying these mutations should be discussed as this is relevant.
One type of study design is proposed in the conclusion, but this has its own challenges and is a similar approach to many of the already included studies. Crucially this type of study design lacks the ability to test change in practice based on test results. Please include discussion around how it may be possible to incorporate more high-quality study designs in this area.
Minor comments
Please introduce genes and their function (e.g. MBOAT etc paragraph line 152) as you have for PNPLA3 etc. previously for the non-specialist reader.
When discussing circulating biomolecules I would begin with the use of AFP as a long term marker of HCC. This is well accepted and more recently elevations of AFP have been shown to predate HCC doi: 10.1016/j.cgh.2020.04.084.
Under molecular biomarkers, line 223, ARID1A is mentioned to contribute to hepatocyte fitness and is implied to be protective from cancer. The function of ARID1A is very much context dependent and is a driver of HCC too. More discussion is required to clarify this.
The list of liver diseases in lines 31-33 includes those aetiologies provided but is not limited to them.
Line 206 – A range of odds ratios is reported, but it is very unclear as to what this refers to and what effect is being measured. The reported ORs span either side of 1 and could be protective/ predictive of HCC – but it is unclear what is being discussed.
I would make it clear that it is serum AFP Line 41.
The poor screening accuracy of both ultrasound and AFP should be introduced.
I would suggest citing additional primary research for line 138 doi.org/10.1073/pnas.1901974116
In the tables – lots of ‘n.a.’ does this mean not applicable or not reported? Important distinction.
Table 1 legend – “Prospective-retrospective” indicates retrospective analysis of prospectively collected cohort in the past. Analysis can’t be prospectively done – rephrase to ‘registry’ if this is correct.
Lines 160 “improve” - I would suggest changing this to worsen. Perhaps the authors could talk about synergy between the risk associated alleles.
Typographic errors are present and require detailed proof reading. Some (non exhaustive) are highlighted below.
Line 31 HCC develops in chronically
Line 37 are no therapies
Line 57 In this review article,
Line 149 Homozygous minor allele of the
Line 233 the challenge, an
Author Response
Point-by-point responses to the reviewers’ comments
We greatly appreciate the thoughtful and insightful comments and suggestions from the reviewers, which help improve clarity and informativeness of this review article. Please see below our point-by-point responses to the comments. Revised parts are underlined in the text and the revised version of the manuscript is attached.
REVIEWER 1
REVIEWER 1, General comment:
This is an important topic and a generally well written manuscript. As the authors highlight hepatocellular carcinoma (HCC) surveillance is an area of debatable cost efficiency and improving enrollment and tailoring patient specific surveillance has the opportunity to optimise cost efficiency and outcomes. The authors have taken a large topic area and aimed describe important studies in the field. It is not clear however how these studies were identified; a systemic approach would be optimal.
Author Reply:
We were based on our previous review articles on this topic (J Hepatol 68;526,2018, Semin Liver Dis 39;153,2019 and Hepatol Res 50;817,2020), and further searched the PubMed and Embase databases with the following search terms to identify more recent publications and possibly missing articles from our precious reviews: hepatocellular carcinoma, screening, surveillance, biomarker, risk prediction, risk score, cirrhosis, and/or prognostic prediction. Among identified studies, we included studies reporting HCC risk scores and biomarkers that were validated in independent patient cohort(s), and included clinical risk scores only when they were externally validated. We have clarified the strategy of inclusion based on validation status in the main text as below (line 82-4). We will follow the journal’s format regarding the strategy of article search in the databases.
We have included HCC risk scores and biomarkers that were validated in independent patient cohort(s), and included clinical risk scores only when they were externally validated.
Overall I enjoyed the review and particularly found tables 1/2 a useful resource and reference. They are however incomplete and this would be aided by a systematic approach. I would highlight a couple of examples where studies are not included in the table which would suggest they have not been detected by the approach taken by the authors. For example personally I use the HCC risk score available at HCCrisk.com which is formed from these two papers both I would recommend including in table 1. DOI: 10.1016/j.jhep.2018.07.024, DOI: 10.1016/j.jhep.2019.05.008. I would also suggest adding the study by Poynard et al. doi: 10.1111/apt.15082 to the list in Table 1. Clarification of the approach taken by the authors for this work should be provided.
Generally, the text is a somewhat superficial overview of the field and would benefit from expanding in some areas (examples given below). In general the discussion of the topic is rather descriptive and doesn’t not give any general advice on relative benefits/pitfalls of the various scores. Exhaustive discussion and clinical recommendations may be beyond the scope of this article however.
Author Reply:
We appreciate the reviewer’s suggestion. We included description about the studies in the main text as below (line 88-91).
In a large VA patient population, HCC-risk-predictive algorithms were developed according to specific clinical contexts, i.e., HCV infection (pre- and post-antiviral treatment), alcoholic cirrhosis, and NAFLD cirrhosis, and implemented in a publicly available web application, HCC risk calculator (hccrisk.com) (J Hepatol 71;523,2019, J Hepatol 69;1088,2018, Aliment Pharmacol Ther 49;308,2019).
REVIEWER 1, Major comments #1:
With conditions that underlie HCC risk continually evolving (i.e. advent of mass Hepatitis B vaccination programmes, effective Hepatitis C therapies and the epidemiological shift toward NAFLD), it is likely that the multiple HCC risk prediction models will evolve in terms of their calibration. Similarly, many of these were developed in specific populations (i.e. east Asian or western Europe or the Americas) where genetic determinants and underlying risk factors for HCC are likely to be substantially different. This is not addressed and is important to ensure equitable and effective surveillance for at risk groups.
Author reply:
We appreciate the reviewer’s suggestion and have added the following description in the Discussion section (line 95-105).
Studies have suggested that specific clinical contexts defined by several factors likely affect accuracy of HCC risk prediction. For example, the ever-evolving antiviral therapies for HBV and HCV will substantially alter the baseline HCC risk level depending on the status of viral control (J Hepatol 21;S0168,2020, Gastroenterol Hepatol 34;436,2019, J Hepatol 64;800,2016). In addition, as suggested from multiple studies on association of germline genotypes with HCC risk, it is plausible that some of such HCC-risk-associated genotypes are bound to patient race/ethnicity, and may guide tailored application of HCC risk prediction by geographic representation of racial/ethnic background. Furthermore, dietary habit and/or food contamination with carcinogen such as aflatoxin could be linked to certain geographic regions, which may allow region-tailored strategy of HCC risk assessment (Clin Liver Dis 24;535,2020). Proper consideration for these factors and incorporation in the risk prediction algorithm may improve accuracy of HCC risk prediction and enhance cost-effectiveness of HCC screening tailored by these parameters.
REVIEWER 1, Major comment #2:
There is mention of machine learning throughout, but no real insight provided into the methods used, how the models were validated and the development process used. It is important these are addressed, as there are known issues with machine learning / artificial intelligence producing models that discriminate against certain population groups (i.e. women/ ethnicities). How can clinical scores that use these approaches become more ethical and equitable?
Author Reply:
We appreciate the insightful comment. We have added the following in the clinical score section (line 109-15).
Despite the promise of recently emerging utilization of machine learning and artificial intelligence approach to develop risk-predictive models, several limitations are worth noting. First, the methodology such as multi-layer neural network is prone to overfit data structure of specific training dataset, which may diminish generalizability of the model (Jama 318;517,2017). In addition, it is possible that the modeling disregard certain patient subgroups depending on the structure of the training set. To mitigate these concerns, it will be increasingly important to ensure transparency of the modeling process and clarification of potential pitfalls (Nature 559;324,2018, NPJ Digit Med 2; 69,2019).
REVIEWER 1, Major comment #3:
The conclusion proposes that HCC risk scores/ biomarkers should be based on full evaluation and performance. However, the review does not touch on the quality of evaluation or the biases contained in many of these studies. Similarly, there is no mention of perhaps the most important aspects to evaluate in any population-based approach – cost and patient acceptability. Some additional discussion around what sorts of testing would be clinically cost effective and acceptable for patients would be beneficial.
Author Reply:
We agree with the important points. We added the following to address the concerns (line 305-17).
Before clinical application of the clinical risk scores and biomarkers, their performance should be fully ascertained on two aspects. First, when a score/biomarker relies on certain devices such as imaging modality or biochemical test, their technical validity should be assessed for reproducibility of the measurements across test dates and sites based on certain technical implementation that will be used in actual clinical setting. Second, magnitude of risk association should be assessed in independent patient series with in-depth consideration on patient demographics and characteristics to evaluate their effects on the performance. In assessing real-world cost-effectiveness of a risk-predictive approach, it is important to consider patient/physician acceptability and adherence particularly because it is known that HCC screening utilization rate is low due to a variety of patient- and practitioner-related reasons (Hepatology 11,2020{ DOI: 10.1002/hep.31309}). This can be estimated by incorporating HCC screening utilization rate in the simulation model as performed in a recent Markov-model-based cost-effectiveness analysis (Clin Transl Gastroenterol 8;e101,2017). In addition, such modeling approach informs desired performance of risk score or biomarker to enable cost-effective HCC screening.
REVIEWER 1, Major comment #4:
I would also suggest the addition of potential alternative strategies for screening based upon stratification. Such a use of cheaper/more expensive methods for stratified patients rather than just altering the frequency. This is, as far as I am aware, an evidence free zone, but conceivably there may be a role for more targeting surveillance with for example cfDNA, GALAD, the use of dynamic biomarker analysis in high risk patients (e.g. DOI: 10.1111/biom.12717 and doi: 10.1371/journal.pone.0156801), or potentially taking patients at high risk clinical score and then using transcriptomics to further refine their risk scores. This area could be discussed.
Author Reply:
We fully agree with the reviewer. We have added the following statement accordingly (line 53-62).
HCC risk stratification may inform tailored HCC screening strategy to maximize cost-effectiveness of HCC screening by optimizing intensity of screening tests according to predicted risk (J Hepatol 71;523,2019, J Hepatol 69;1088,2018, Aliment Pharmacol Ther 49;308,2019). For example, more frequent HCC screening can be offered to high-risk patients compared to low-risk patients. It may be justifiable to utilize costly but high-performance screening tests such as advanced imaging modalities (e.g., MRI-based examination) and biomarkers (e.g., circulating cell-free methylated DNA and GALAD score) in high-risk patients. Given the limited resources for HCC screening in real-world clinical practice, prioritizing high-risk patients for regular HCC screening will also be a rational approach. This should be a focus of future research in HCC screening and early detection. A modeling-based study showed that this personalized strategy indeed can be a viable approach to enable cost-effective HCC screening (Clin Transl Gastroenterol 8;e101,2017).
REVIEWER 1, Major comment #5:
The trials for altering screening frequency from 3-12 months were in an unstratified population but pragmatically six months makes sense in relation to tumor doubling time; see http://dx.doi.org/10.1136/gutjnl-2020-321040. Therefore, I would caution the suggestion for less frequent surveillance as this may just lead to late stage detection and negate any benefit of performing surveillance at all. In support of this there is some emerging data to suggest that poor adherence to surveillance is associated with equivalent outcomes to no surveillance at all. Similarly, I would be interested in the authors’ opinion on what would be deemed sufficient from any of these risk scores to remove a patient from ongoing surveillance completely. I would however discourage the provision of clinical advice within this article and leave this area to the guidelines of the major associations.
As the annual risk of developing HCC is relatively low, the limiting factor for uptake of HCC surveillance in practice is cost (to either healthcare system or patients). With the burdens of disease in low-middle incomes shifting to non-communicable diseases, for example, how could these be made applicable for future generations of patients at risk of HCC?
Author Reply:
We agree with the reviewer’s comment. We have added the following (line 64-74).
With the currently recommended HCC screening based on ultrasound in the all-comer setting, alteration of HCC screening frequency did not influence patient outcome in clinical studies (Gut 2020{DOI: 10.1136/gutjnl-2020-321040}). This issue may be carefully revisited based on these experiences when studying new HCC screening modalities with improved performance with consideration about anticipated tumor growth rate. Sparing HCC screening in low-risk patients could substantially mitigate the burden of regularly screening the large patient population at risk. However, such decision of dropping a subset of patients from regular screening should be carefully made to minimize risk of late tumor diagnosis, which may incur increased medical care costs despite poor prognosis. Furthermore, with the shift of major HCC etiology from communicable viral infection to metabolic disorders accompanied with low HCC incidence rate and disproportionally affecting the communities with low socio-economic status, outreach effort to the at-risk population will become increasingly important.
REVIEWER 1, Major comment #6:
In the transcriptome section, the study flits between discussing methods of early detection in patients at risk of HCC (i.e. populations with no HCC, but with preexisting liver disease) and those who have had HCC and markers of post surgical recurrence. These are two very different groups and two very different clinical scenarios. For example, at the population level obtaining liver biopsies on any patient at risk of HCC is not feasible and posses additional risk to patients. Whilst performing transcriptomic on resection tissue is relatively simpler but still requires specific clinical pathways and has cost/resource implications.
Similarly embarking upon transcriptomic signatures is more expensive and reliant on liver tissue (as highlighted above). The relevance between resection based transcriptome and a smaller biopsy sample should be discussed. Similarly the risk of performing a biopsy in a patient purely to obtain a cancer risk transcriptome should be weighed against the risk of the biopsy itself if not being performed for other clinical indications.
Author Reply:
We appreciate the reviewer’s thoughtful comments. We have discussed this important issue as below (line 220-9).
Hepatic tissue transcriptome profiling has been widely used as a reliable resource to explore biological dysregulation associated with patient prognosis and clinical phenotypes. Earlier studies analyzed diseased liver tissues obtained at the time of hepatic surgery performed as treatment of liver tumor (Hepatol Res 50;817,2020). These studies demonstrated the proof of concept that hepatic transcriptomic dysregulation is associated with the risk of developing de novo HCC after the surgical therapies (N Engl J Med 359;1995,2008). However, the HCC-risk-predictive performance should be re-evaluated in HCC-naïve patients for prediction of future HCC risk. In addition, benefit of performing liver biopsy should be justified over the potential risk of adverse events caused by the procedure such as bleeding. Thus, surrogate markers of hepatic transcriptome will overcome the limitation and enable clinically applicable HCC risk prediction with enhanced accuracy based on molecular information.
REVIEWER 1, Major comment #7:
I also feel that a discussion regarding the now appreciated clonality of cirrhotic nodules and the present of cancer associated mutations in the tissue are relevant here doi: 10.1016/j.cell.2019.03.026 and doi: 10.1038/s41586-019-1670-9. Relevant review (e.g. doi: 10.1016/j.jhep.2020.01.019).
The roles of mutational signatures or mutational load in liver tissue is not discussed, just bulk transcriptomics. These mutational signatures could be used to assist in defining aetiology as well as HCC risk. This could be included and discussed briefly but is understandably outwith the scope of clinical practice currently.
Author Reply:
We appreciate the suggestion and we also think this point is important in this review. We have added a new subsection as below (line 252-64).
3.3. Somatic DNA mutations in non-malignant liver
Somatic DNA mutations are in general implicated as oncogenic drives in a variety of cancer types, including HCC. Recent studies have suggested that more somatic DNA mutations also accumulate in cirrhotic livers compared to healthy livers and possibly affect propensity of carcinogenic, and could be detected in liver tissues and in circulation (J Hepatol 72;990,2020, Cell 177;608,2019, Nature 574;538,2019, Hepatol Commun 2;718,2018). In addition, a recent study suggested that somatic mutations in PKD1, KMT2D, and ARID1A genes in non-malignant cirrhotic liver protect hepatocytes from malignant transformation (Hepatocellular Carcinoma: Translational Precision Medicine Approaches, Hoshida,Y.,Ed). Of note, it is noteworthy that the same somatic events such as dysregulation of ARID1A gene can either promote or suppress HCC initiation and progression depending on the stage of hepatocarcinogenesis. In addition, rodent-model-based studies have suggested that hepatocyte polyploidy confers protection from HCC development, which can be therapeutically induced by modulation of a cytokinesis gene/protein, anillin (Gastroenterology 158;1698,2020). To be considered as biomarkers for HCC risk prediction, detectability in clinically accessible specimens will be a key issue for this type of molecular information.
REVIEWER 1, Major comment #8:
I would also be explicit for the general reader that because the mutations being discussed in section 3.1 are germline this could be tested once to inform lifetime risk. The costs of assaying these mutations should be discussed as this is relevant.
Author Reply:
We appreciate the suggestion, and added the following statement (line 143-5).
Germline SNPs can be easily assessed using buccal swab or peripheral blood sample at any time point because they do not change throughout life, and are increasingly more accessible with decreasing costs over time as a viable tool for potential molecular HCC risk prediction.
REVIEWER 1, Major comment #9:
One type of study design is proposed in the conclusion, but this has its own challenges and is a similar approach to many of the already included studies. Crucially this type of study design lacks the ability to test change in practice based on test results. Please include discussion around how it may be possible to incorporate more high-quality study designs in this area.
Author Reply:
We echo the reviewer’s concern. We have discussed this issue as below (line 321-5).
These expanding resources following the PRoBE design are expected to provide more reliable measures of effect size in risk estimation, which will inform design and required sample size in subsequent interventional studies to examine magnitude of clinical benefit from the risk-stratified approach. Such evidence will eventually support decision whether the tailored strategy can be incorporated into clinical practice.
REVIEWER 1, Minor comment #1:
Please introduce genes and their function (e.g. MBOAT etc paragraph line 152) as you have for PNPLA3 etc. previously for the non-specialist reader.
Author Reply:
We have added explanations for several genes that has some relative functions to liver diseases accordingly.
REVIEWER 1, Minor comment #2:
When discussing circulating biomolecules I would begin with the use of AFP as a long term marker of HCC. This is well accepted and more recently elevations of AFP have been shown to predate HCC doi: 10.1016/j.cgh.2020.04.084.
Author Reply:
We thank you for your suggestions. I have added the description about AFP as a risk stratification marker as follows (line 266-8):
AFP has been reported as a biomarker for not only an early HCC detection but a risk stratification. Hughes et al. revealed that a longitudinal change in AFP can more accurately capture future HCC risk (Clin Gastroenterol Hepatol,2020).
REVIEWER 1, Minor comment #3:
Under molecular biomarkers, line 223, ARID1A is mentioned to contribute to hepatocyte fitness and is implied to be protective from cancer. The function of ARID1A is very much context dependent and is a driver of HCC too. More discussion is required to clarify this.
Author Reply:
Thank for pointing out the important point. We have added the following description (line 258-60).
Of note, it is noteworthy that the same somatic events such as dysregulation of ARID1A gene can either promote or suppress HCC initiation and progression depending on the stage of hepatocarcinogenesis.
REVIEWER 1, Minor comment #4:
The list of liver diseases in lines 31-33 includes those aetiologies provided but is not limited to them.
Author Reply:
We have replaced “i.e.” with “e.g.” accordingly.
REVIEWER 1, Minor comment #5:
Line 206 – A range of odds ratios is reported, but it is very unclear as to what this refers to and what effect is being measured. The reported ORs span either side of 1 and could be protective/ predictive of HCC – but it is unclear what is being discussed.
Author Reply:
We have corrected the description as follows (line 278).
(ORs for high-risk and low-risk metabolites, 2.16-6.78 and 0.27-0.56, respectively, for incremental one standard deviation)
REVIEWER 1, Minor comment #6:
I would make it clear that it is serum AFP Line 41.
Author Reply:
We have corrected accordingly.
REVIEWER 1, Minor comment #7:
The poor screening accuracy of both ultrasound and AFP should be introduced.
Author Reply:
We added the description about this in the added paragraph in the molecular section (line 43).
diagnostic accuracy of ultrasound and AFP is suboptimal
REVIEWER 1, Minor comment #8:
I would suggest citing additional primary research for line 138 doi.org/10.1073/pnas.1901974116
Author Reply
We have added the following description underlined (line 182-3).
The PNPLA3 I148M variant causes impaired triglyceride mobilization and accumulation of lipid droplet by evading ubiquitylation and by comparative gene identification‐58 (CGI‐58)‐dependent inhibition of adipose triglyceride lipase, resulting in hepatic steatosis (Proc Natl Acad Sci U S A 116; 9521,2019).
REVIEWER 1, Minor comment #9:
In the tables – lots of ‘n.a.’ does this mean not applicable or not reported? Important distinction.
Author Reply:
We have changed the description of “n.a. (not available/applicable)” to “n.a. (not applicable) or n.r. (not reported)” in the table 1 and 2.
REVIEWER 1, Minor comment #10:
Table 1 legend – “Prospective-retrospective” indicates retrospective analysis of prospectively collected cohort in the past. Analysis can’t be prospectively done – rephrase to ‘registry’ if this is correct.
Author Reply:
The header “assessment” was replaced with “registry” accordingly.
REVIEWER 1, Minor comment #11:
Lines 160 “improve” - I would suggest changing this to worsen. Perhaps the authors could talk about synergy between the risk associated alleles.
Author Reply:
We have corrected accordingly.
REVIEWER 1, Minor comment #12:
Typographic errors are present and require detailed proof reading. Some (non exhaustive) are highlighted below.
Line 31 HCC develops in chronically
Line 37 are no therapies
Line 57 In this review article,
Line 149 Homozygous minor allele of the
Line 233 the challenge, an
Author Reply:
We have corrected all accordingly.

Reviewer 2 Report
The article by Dr Kubota et al is an overview of clinical risk scores, germline DNA variants, transcriptomic signatures of liver tissue, and circulating biomolecules, as risk scores for HCC development in patients with chronic liver disease. The paper is well-written, easy to follow and comprehensive regarding the clinical risk scores, DNA variants and transcriptomic profiles. The description of circulating biomarkers could be somewhat extended (see comments below).
Specific comments:
- The description of screening strategies in high-, intermediate, and low-risk patients (page 2 lines 46-49, and figure 1) is slightly inexplicit. One may get the impression that low-risk patients should undergo screening less frequently than high-risk patients. However, the screening interval is not determined by the risk of HCC development (i.e. the HCC incidence in the specified risk group) but rather by the tumor doubling time. The HCC incidence instead determines whether or not screening should be performed at all. This decision is based on cost-effectiveness analyses and a common threshold is an annual HCC incidence of 1.5 %. I think the authors mean this, but the text on line 46-49 is a little bit confusing and could be rephrased. Also, in Figure 1, the terms “more intensive screening” and “less intensive screening” could be formulated more exact to gain clarity, since low-risk patients probably should not be screened at all, as is pointed out in reference 12.
- The paragraph 3.3 “Circulating biomolecules” is a nice review of biomarkers associated to increased HCC risk. However, the 28-protein panel, reference no. 153 (described on lines 207-211) is merely a diagnostic biomarker of HCC. This part is clearly of interest, however if biomarkers of early diagnosis also should be included in this review, I would suggest a separate short paragraph on this topic mentioning other commonly used protein biomarkers (i.e. AFP-L3, PIVKA-II, glypican-3 and Golgi protein 73), including recent scores for early HCC detection (e.g. GALAD score).
Minor comments:
- Page 1, line 31: Missing an “in” (HCC develops … in.. chronically)
- Page 1 line 40: ”diagnoses..” should be “diagnosed..”
- Page 2 line 57: “This is review article..” should be “In this review article..”
Author Response
Point-by-point responses to the reviewers’ comments
We greatly appreciate the thoughtful and insightful comments and suggestions from the reviewers, which help improve clarity and informativeness of this review article. Please see below our point-by-point responses to the comments. Revised parts are underlined in the text and the revised version of the manuscript is attached.
REVIEWER 2
REVIEWER 2, Major comment 1:
The description of screening strategies in high-, intermediate, and low-risk patients (page 2 lines 46-49, and figure 1) is slightly inexplicit. One may get the impression that low-risk patients should undergo screening less frequently than high-risk patients. However, the screening interval is not determined by the risk of HCC development (i.e. the HCC incidence in the specified risk group) but rather by the tumor doubling time. The HCC incidence instead determines whether or not screening should be performed at all. This decision is based on cost-effectiveness analyses and a common threshold is an annual HCC incidence of 1.5 %. I think the authors mean this, but the text on line 46-49 is a little bit confusing and could be rephrased. Also, in Figure 1, the terms “more intensive screening” and “less intensive screening” could be formulated more exact to gain clarity, since low-risk patients probably should not be screened at all, as is pointed out in reference 12.
Author Reply:
Thank you for your suggestions to make our manuscript clearer. I have simplified the description on line 47-8 to avoid the confusion as follows:
This approach leads to over-screening of low-risk patients and under-screening of high-risk patients (Hepatology,2020{DOI: 10.1002/hep.31309})
We have also modified the descriptions of the figure 1.
We have rephrased “Less intensive screening” to “Less intensive screening or Free from screening” (Please see the attachment).
REVIEWER 2, Major comment 2:
The paragraph 3.3 “Circulating biomolecules” is a nice review of biomarkers associated to increased HCC risk. However, the 28-protein panel, reference no. 153 (described on lines 207-211) is merely a diagnostic biomarker of HCC. This part is clearly of interest, however if biomarkers of early diagnosis also should be included in this review, I would suggest a separate short paragraph on this topic mentioning other commonly used protein biomarkers (i.e. AFP-L3, PIVKA-II, glypican-3 and Golgi protein 73), including recent scores for early HCC detection (e.g. GALAD score).
Author Reply:
We really appreciate the reviewer’s comments. As the reviewer pointed out, the main topic of this article is future HCC risk prediction, so we removed the description about the 28-protein panel to avoid the confusion.
REVIEWER 2, Minor comments:
- Page 1, line 31: Missing an “in” (HCC develops … in.. chronically)
- Page 1 line 40: ”diagnoses..” should be “diagnosed..”
- Page 2 line 57: “This is review article..” should be “In this review article..”
We have corrected accordingly.
